# Conserved molecular signatures of hygrosensory neurons in two dipteran species

Kristina Corthals[1¤*], Ganesh Giri[1], Johan Reimegård[2], Allison Churcher[3], Anders Enjin[1]

**1** Department of Experimental Medical Science, Lund University, Lund, Sweden, **2** Department of Cell and Molecular Biology, National Bioinformatics Infrastructure Sweden, Science for Life Laboratory, Uppsala University, Uppsala, Sweden, **3** Department of Plant Physiology, National Bioinformatics Infrastructure Sweden, Science for Life Laboratory, Umeå University, Umeå, Sweden

¤ Current address: Institute for Infection Prevention and Control, Medical Faculty, University of Freiburg, Freiburg, Germany

* kristina.corthals@med.lu.se

## Abstract

Small poikilothermic animals like insects rely on environmental sensing for survival. The ability to detect humidity through specialized sensory neurons is particularly critical, allowing them to maintain water balance across diverse environments. While recent studies have identified key receptors associated with humidity sensing, our understanding of the underlying molecular architecture of these sensory systems remains incomplete. Here, we conducted a comparative analysis of single-nucleus transcriptomes of humidity receptor neurons (HRNs) between the vinegar fly *Drosophila melanogaster* and the yellow fever mosquito *Aedes aegypti*. We identified 21 shared genes that contribute to the molecular identity of HRNs in both species. These genes encode proteins involved in transcriptional regulation, cellular signalling, enzymatic pathways and cellular organization. Through behavioural analyses, we demonstrate that two of these genes, the serotonin receptor *5-HT7* and the kinesin motor protein *Kif19A*, are both necessary for humidity-guided behaviours in adult flies. The conservation of these genes between species separated by over 150 million years of evolution suggests shared functional requirements for humidity sensing in dipterans. Our findings provide insights into fundamental principles of sensory neuron organization and offer a framework for understanding how specialized sensory systems evolve and maintain their function.

## Introduction

Environmental humidity shapes terrestrial ecosystems and influences animal behaviour and survival [1,2]. For insects, as small poikilothermic animals, the ability to detect and respond to environmental moisture levels is particularly important. It affects their distribution, survival, and ecological interactions across diverse habitats

**Data availability statement:** Skripts for analysis done in the paper can be found here: https://github.com/hygrosensation/ComparativeStudy Details on the humidity arena: Giri, G., Nagloo, N. & Enjin, A. A dynamic humidity arena to explore humidity-related behaviours in insects. Journal of Experimental Biology 227, jeb.247195 (2024). Drosophila Dataset: https://cloud.flycellatlas.org/index.php/s/PcHBoL23CmxGNsb Li, H. et al. Fly Cell Atlas: a single-nucleus transcriptomic atlas of the adult fruit fly. Science 375, eabk2432 (2022). Aedes aegypti datasets: Adavi, E. D. et al. Olfactory receptor coexpression and co-option in the dengue mosquito. 2024.08.21.608847 Preprint at https://doi.org/10.1101/2024.08.21.608847 (2024). Herre, M. et al. Non-canonical odor coding in the mosquito. Cell 185, 3104-3123.e28 (2022).

**Funding:** Open access funding provided by Lund University. The funding was provided by Wenner-Gren Stiftelserna, Svenska Forskningsrådet Formas (2021-02008), Knut och Alice Wallenbergs Stiftelse, Vetenskapsrådet (202103772), Crafoordska Stiftelsen, Jeanssons Stiftelser. KC was financially supported by a Formas Mobility Grant (2021-02008). AC and JR were financially supported by the SciLifeLab & Wallenberg Data Driven Life Science Program, Knut and Alice Wallenberg Foundation (KAW 2020.0239, KAW 2017.0003), and by the National Bioinformatics Infrastructure Sweden (NBIS) at SciLifeLab. The funders had no role in study design, data collection and analysis, decision to publish, or preparation of the manuscript.

**Competing interests:** The authors have declared that no competing interests exist.

[3,4]. The capacity to sense humidity serves multiple functions across insect species. Pollinators such as the bumble bee *Bombus terrestris*, the white-lined sphinx moth *Hyles lineata* and the tobacco hawk moth *Manduca sexta* use floral humidity gradients to locate and identify flowers [5–7]. In disease vectors like the yellow fever mosquito *Aedes aegypti* and the malaria mosquito *Anopheles gambiae*, humidity sensing plays a crucial role in both host-seeking and oviposition site selection [8–10].

Insects detect humidity through specialized sensory structures called hygrosensilla which are located on their antennae [11]. Each hygrosensillum houses a characteristic group of three humidity receptor neurons (HRNs), known as a hygrosensory triad. This triad consists of three functionally distinct neurons: a moist neuron that is activated when humidity increases, a dry neuron that responds when humidity decreases and a hygrocool neuron that activates when temperature drops [12]. The organization of these hygrosensilla varies among insect species. In *D. melanogaster*, multiple hygrosensilla are clustered together within an invagination on the posterior side of the antenna called the sacculus [13]. In contrast, in *Ae. aegypti*, individual hygrosensory triads are housed in separate invaginations, each connected to the antennal surface through a small pore [14].

The molecular basis of humidity and temperature sensing in insects involves several members of the ionotropic receptor (IR) gene family. In *D. melanogaster*, *Ir40a* is expressed in dry-responsive neurons in both chamber I and II of the sacculus, while *Ir68a* is found in moist-responsive neurons [15–18]. Hygrocool cells show a chamber-specific expression pattern, with *Ir21a* expressed in chamber I and *Ir40a* in chamber II [19]. Additionally, *Ir25a* and *Ir93a* are expressed in all these sensory neurons. Behavioural studies using mutants have shown that loss of either *Ir25a* or *Ir93a* disrupts both humidity- and temperature-guided behaviours [15,17,20]. *Ir40a* and *Ir68a* mutants have impaired humidity sensing, while *Ir21a* mutant flies show normal humidity responses, despite being expressed in HRNs [15–19,21].

These ionotropic receptor-mediated mechanisms are conserved in mosquitoes. Recent studies have shown that *Ir93a* is essential for both temperature and humidity sensing in mosquitoes [10]. Similar to *D. melanogaster*, *Ir21a* mutant mosquitoes show specific defects in temperature sensing while maintaining normal humidity responses, while *Ir40a* is associated with dry air detection and *Ir68a* with humid air detection [14,22].

Across these species, the ionotropic receptors *Ir93a*, *Ir40a*, and *Ir68a* are critical molecular components of hygrosensation, though their precise role in the transduction mechanism remains to be established. Additional molecular components have been implicated in humidity-related behaviours, including the TRP channels *water witch* and *nanchung*, although their expression in HRNs has not been demonstrated and their contribution to hygrosensation remains to be clarified [15,17,23]. The odorant binding protein *Obp59a* localizes to support cells of hygrosensory sensilla rather than the HRNs themselves, suggesting that the molecular apparatus underlying humidity detection extends beyond the sensory neurons [24]. However, beyond these candidates, little is known about the broader molecular architecture that enables hygrosensory neurons to develop and function. In a previous study, we identified

distinct transcriptional profiles of hygrosensory neurons in the sacculus of *D. melanogaster*, revealing unique molecular signatures for dry, moist and hygrocool neurons, along with specialized support cells [25]. Here, we build upon these insights and present comparative transcriptomic analysis of hygrosensory neurons in *D. melanogaster* and *Ae. aegypti*, two species separated by over 150 million years of evolution, to identify a conserved molecular toolkit of 21 genes that are expressed in these specialized neurons. Among these, we demonstrate essential roles for two previously uncharacterized genes in humidity sensing: the serotonin receptor *5-HT7* and the kinesin motor protein *Kif19A*. Our findings reveal fundamental organizational principles of hygrosensory neurons and provide insights into how specialized sensory systems are built and maintained across evolution.

## Materials and methods

### Methods

**Single-nucleus transcriptome analysis.** *Data:* To analyze the transcriptomic profiles of hygrosensory (HRN) populations in *D. melanogaster* antennae, we utilized the mixed-sex 10x Genomics antennal dataset from the Fly Cell Atlas [26]. For *Ae. aegypti*, we used two single-nucleus datasets of female antennae [27,28]. Given the potential discrepancies in gene annotations and processing methodologies between the two available *Ae. aegypti* datasets, we opted against their integration to preserve the integrity of our downstream analyses. This decision mitigates the risk of introducing biases or artifacts that could arise from merging datasets with unknown preprocessing differences. Consequently, we will treat and analyse these datasets independently, ensuring robust and reliable results for each set. Quality control and subsequent analyses were performed using the Seurat package Version 5.1.0 [29].

*Extraction of neuronal cells.* All datasets were previously dimensionally reduced and clustered. The *D. melanogaster* dataset was processed as previously described [25]. Clusters were assigned neuronal identity based on the expression of 4 neuronal marker genes: *Syt1*, *elav*, *CadN*, *brp* [30–33] (S1A Fig) and their *Ae. aegypti* orthologues *LOC5565901* (orthologue to *Syt1*), *LOC5570204* (orthologue to *elav*), *LOC5564848* (orthologue to *CadN*), *LOC5570381* (orthologue to *brp*) (S2A Fig). Additionally, glia cells were classified using the *D. melanogaster* glial marker *repo* and the *Ae. aegypti* orthologue *LOC110678282* (S1B and S2B Figs) [34].

In the Herre et al. *Ae. aegypti* dataset, clusters 3 and 45 were excluded from the subsequent analysis due to their ambiguous neuronal marker expression profiles (S2A-B Fig). While these clusters exhibited minor expression of *Ir25a*, they lacked significant expression of key hygroreceptors, including *Ir93a*, *Ir40a*, and *Ir21a* (S3 Fig). This absence of relevant receptor expression, combined with their unclear neuronal identity, justified their exclusion from further analyses.

Following neuron-specific filtering the datasets comprised: *D. melanogaster* (this study): 15,491 nuclei; *Ae. aegypti* dataset 1 (Herre et al., 2022): 5,175 nuclei; *Ae. aegypti* dataset 2 (Adavi et al., 2024): 46,073 nuclei. Full quality control metrics for the mosquito datasets are reported in the original publications [27,28].

*Reprocessing of neuronal clusters and dimensionality reduction:* The neuronal clusters were extracted and reprocessed via normalization, PCA analysis and Nearest-Neighbor Clustering. The *D. melanogaster* dataset was re-clustered using 40 principal components (PCs) with a resolution of 0.5. The Herre et al. *Ae. aegypti* dataset was analyzed using 40 PCs with a resolution of 1.0. Dimensionality reduction was performed by using Uniform Manifold Approximation and Projection (UMAP).

*Identification of clusters of interest:* To identify hygrosensory neuron clusters in both species, we used expression of previously described ionotropic receptors. In *D. melanogaster*, we examined *Ir25a*, *Ir93a*, *Ir40a* and *Ir21a* (*Ir68a* expression was negligible, consistent with previous studies [25]). In *Ae. aegypti*, we examined *Ir93a*, *Ir21a* and *Ir40a* (*Ir68a* was not significantly detected). We then extracted the top 5 and top 50 marker genes using Seurat FindMarkers for the resulting clusters of interest.

*Identification of overlapping genes:* To identify genes common to HRNs in both species, we compared the top 50 marker genes from the identified clusters of interest in *D. melanogaster* (clusters 14, 27) with those in *Ae. aegypti* (cluster 24/cluster 39, 41). To ensure a comprehensive and comparative analysis, we used OrthoDB to identify orthologous genes

in both *D. melanogaster* and *Ae. aegypti* for the top 50 marker genes (MG) of each cluster [35]. The resulting lists were then compared as follows:

1. Top 50 MG (cluster 14, 27) from *D. melanogaster* x Top 50 MG (cluster 24/cluster 39, 41) from *Ae. aegypti* as *D. melanogaster* orthologues

2. Top 50 MG (cluster 24/cluster 39, 41) from *Ae. aegypti* x Top 50 MG (cluster 14, 27) from *D. melanogaster* as *Ae. aegypti* orthologues

This resulted in the identification of a set of genes that are present in the Top 50 marker genes in both the *D. melanogaster* and the *Ae. aegypti* hygrosensory clusters.

**Behavioural analysis *D. melanogaster*.** *Animal husbandry and handling:* D. melanogaster strains were maintained at 25°C on cornmeal agar food under 12-hour dark/light cycle within an incubator. Humidity levels inside the vials ranged from 70% RH to 90% RH depending on distance to the food source. Prior to the experiment individual *D. melanogaster* were anesthetized on ice and a metal pin was fixed on their thorax using light curing glue (Heliobond, Levitat Limited, United Kingdom). Subsequently they were starved and desiccated at 5% RH and 21–22 °C for 4 h. Both male and female flies were used in experiments.

*Fly strains:* D. melanogaster strains were obtained from Bloomington Stock Center with the following IDs: #5905 (*w[1118]*), #84446 (*w[*]; TI{RFP[3xP3.cUa]=TI}5-HT7[attP]*), #56735 (*y[1] w[*]; Mi{y[+mDint2]=MIC}Kif19A [MI12222]*), #17177 (*w[1118]; P{w[+mC]=EP}Ir21a[EP526]*). Mutant lines were not backcrossed to the control strain. All lines carry *white* mutant alleles on otherwise similar genetic backgrounds, reducing the likelihood that background effects account for the observed phenotypes.

*Behavioural data acquisition:* The behavioural data was recorded using the previously described closed-loop dynamic humidity arena [20]. Tethered and desiccated flies were positioned on a 9 mm plastic ball. The movement of the ball in response to the walking pattern was recorded at 150 fps (Basler acA-1920) and the real-time trajectory is reconstructed using FicTrac [36]. The flies were presented with a step humidity stimulus: 10%RH – 80% RH – 10% RH with each phase being maintained for 500 seconds. During the experiment humidity and temperature were continuously monitored. Trials were stopped if there were any tracking failures or errors in stimulus delivery. Incomplete trials were subsequently removed from the analysis.

*Data analysis:* The obtained real-time trajectories of individual *D. melanogaster* in response to the humidity stimulus along with humidity level, temperature and time were extracted and used for subsequent analysis.

For noise removal an exponential weighted moving average filter with a span of 50 was applied. Speed data was obtained using the cumulative Euclidean distance between data points and time interval values. For comparability, speed data was normalized and mean speed calculated for 10% RH and 80% RH, respectively (for a more detailed description see [20]).

*Statistics:* Given the non-normal distribution of the data, statistical significance was evaluated using non-parametric bootstrap resampling (n = 10,000 iterations). For each genotype, the observed difference in mean walking speed was computed across the three relative humidity (RH) setpoints: initial 10% RH, 80% RH, and final 10% RH. Speed measurements across humidity conditions were pooled and resampled with replacement to generate empirical null distributions of bootstrap differences. P-values were derived as the proportion of resampled differences exceeding the observed absolute difference in absolute value. Comparisons were conducted via the Mann-Whitney U test with Bonferroni correction for multiple comparisons.

## Results

### Identification of HRN populations in fly and mosquito antennal transcriptomes

In the *D. melanogaster* antennal transcriptome, we identified two clusters (14, 27) expressing *Ir93a*, *Ir40a* and *Ir21a* that were also negative for markers of olfactory neurons (Fig 1A, S4A). In line with previous single-cell transcriptomic studies,

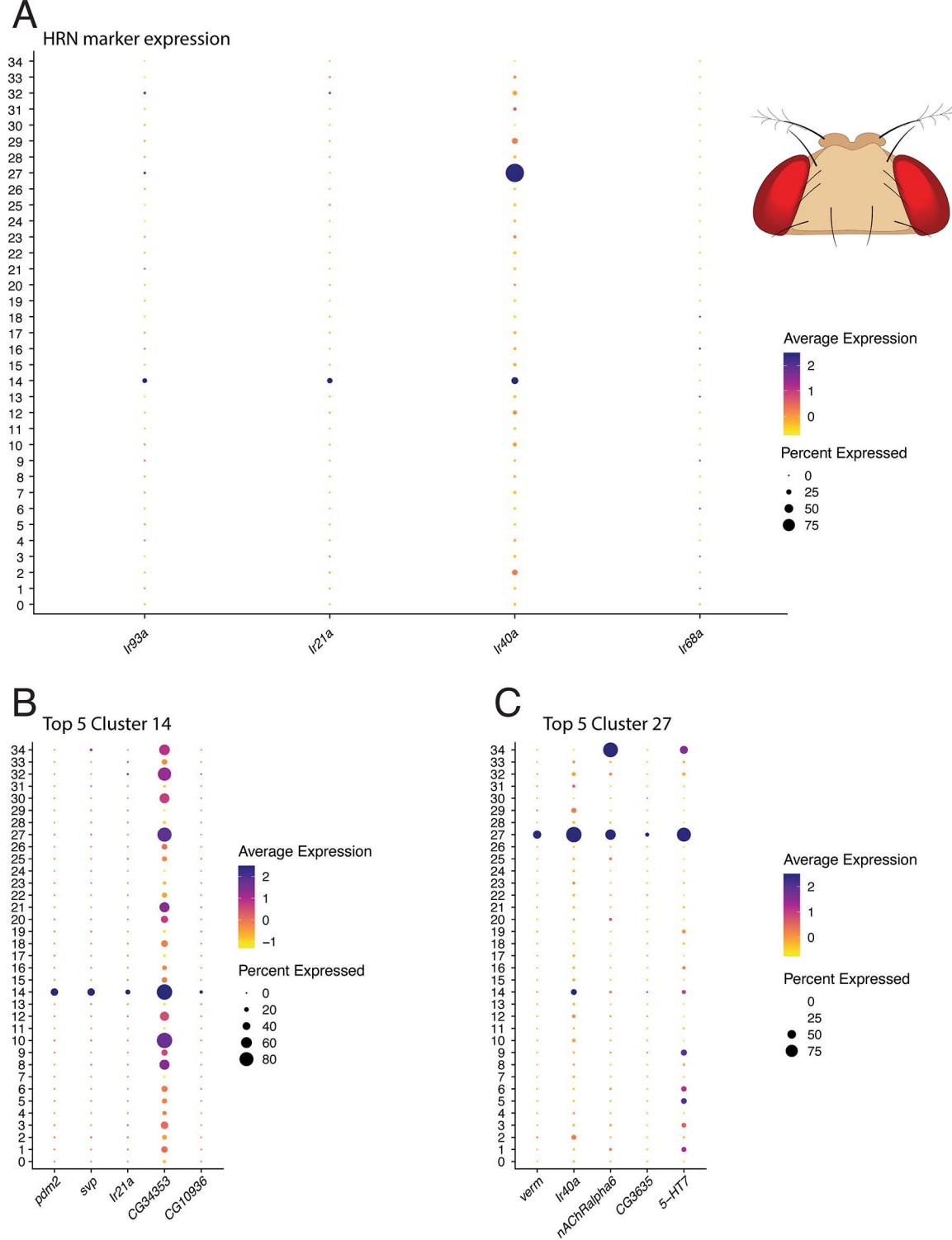

**Fig 1. Two candidate HRN clusters in the _D. melanogaster_ antennal transcriptome co-express canonical hygro- and thermoreceptor markers.** (A) Dot plot showing expression of the ionotropic receptor genes _Ir93a_, _Ir40a_, _Ir68a_ and _Ir21a_ across all antennal neuronal clusters from the Fly Cell Atlas single-nucleus RNA-seq dataset. Clusters 14 and 27 show co-expression of _Ir93a_, _Ir40a_ and _Ir21a_, consistent with a hygrosensory identity.

*Ir68a* expression is negligible throughout, consistent with previous single-nucleus studies [25]. (B–C) Top five marker genes for the two candidate HRN clusters 14 and 27. Dot size represents the proportion of nuclei expressing the gene within that cluster (percent expressed); colour indicates average expression level, ranging from low (yellow) to high (purple).

*Ir68a* expression was negligible and non-specific throughout the antenna [25]. Cross-referencing cluster-specific markers with our earlier transcriptomic analysis of HRNs revealed that cluster 14 comprises hygrocool (marker genes include *Ir21a*, *Ir40a*, *Ir93a*), moist (marker genes include *Ac13E*, *Nlg3*, *fred*), and arista temperature cells (marker genes include *Ir21a*, *Ir93a*, *Pdfr*), based on marker gene expression profiles (Fig 1B, S5A). Cluster 14 was treated as a combined HRN population for the comparative analysis, as it contains multiple HRN subtypes that collectively represent hygrosensory and temperature-responsive neurons of the sacculus and arista. Cluster 27 likely contains dry cells (marker genes include *Ir40a*, *verm*, *nAChRα6*, *5-HT7*; Fig 1C, S5B).

Analysis of the two *Ae. aegypti* antennal transcriptomes revealed expression of *Ir93a*, *Ir21a*, and *Ir40a*, with *Ir40a* showing notably low expression levels compared to previous studies (Fig 2). Consistent with our *D. melanogaster* findings, *Ir68a* expression was insignificant across both datasets. We identified cluster 24 in the Herre et al. dataset (Fig 2A, B) and clusters 39 and 41 in the Adavi et al. dataset (Fig 2C-E) as HRN populations. The presence of *Ir40a*, which is exclusively expressed in HRNs, combined with the consistent expression of *Ir93a* and *Ir21a* across both independent datasets, establishes these clusters as HRNs in the *Ae. aegypti* antenna.

**Conserved genetic profiles of hygrosensory neurons in dipterans.** To determine the conserved genetic profile of HRNs, we compared gene expression profiles between these sensory neurons in *D. melanogaster* and *Ae. aegypti*. By cross-referencing the top 50 marker genes from the HRN clusters in both species (*D. melanogaster* clusters 14, 27; *Ae. aegypti* cluster 24 from Herre, and clusters 39, 41 from Adavi), we identified 21 genes, that likely represent evolutionary conserved components of dipteran humidity sensing (Figs 3A-C, S5, S6).

These conserved genes fall into distinct functional categories. The largest group consists of transcriptional regulators: the steroid receptor seven-up (*svp*), the BTB/POZ domain nuclear factor ribbon (*rib*), LIM homeobox 1 (*Lim1*), the POU/homeodomain transcription factor nubbin (*nub*), the zinc finger transcriptional repressor spalt-related (*salr*), the homeodomain transcription factor homothorax (*hth*) and the C2H2 zinc-finger transcription factor disco-related (*disco-r*). The presence of multiple conserved transcription factors suggests a complex, hierarchical regulation of sensory neuron identity in the sacculus and arista.

A second category includes proteins involved in cellular signalling and ion transport: the serotonin receptor *5-HT7*, the potassium channel *CG42594*, the arrestin *CG32683* and the already known ionotropic receptors *Ir21a*, *Ir40a* and *Ir93a*. We also identified several enzymes: the dopamine beta-monooxygenase *olf413*, involved in catecholamine synthesis; *CG9743*, which shows stearoyl-CoA desaturase activity; and *CG3655*, which functions as a glycosyltransferase.

Finally, we identified structural and adhesion components including Kinesin family member 19A (*Kif19A*), a microtubule-associated motor protein, the GPI-anchored proteins *CG14274*/*witty* and *CG32432*, Down syndrome cell adhesion molecule 4 (*Dscam4*) and friend of echinoid (*fred*) suggesting the importance of specific cellular recognition and organization in these sensory systems.

For a more detailed understanding of the relationship between specific genes and neuronal subtypes, we conducted an expression analysis using the sub-clustered dataset of sacculus neurons and associated support cells, building upon our previous study#39;s methodology [25]. This approach allowed for a more nuanced examination of gene expression patterns within these specialized structures. This analysis revealed distinct expression patterns across various sensory neuron subtypes (Fig 4 A, B). In hygrocool and moist cells, we detected expression of the steroid receptor *svp*. The transcription factor *rib* exhibited differential expression across hygrosensory neuron subtypes, with notably high levels in hygrocool cells of chamber I and lower expression levels in moist cells. We further observed distinct expression of *Lim1* in hygrocool, moist,

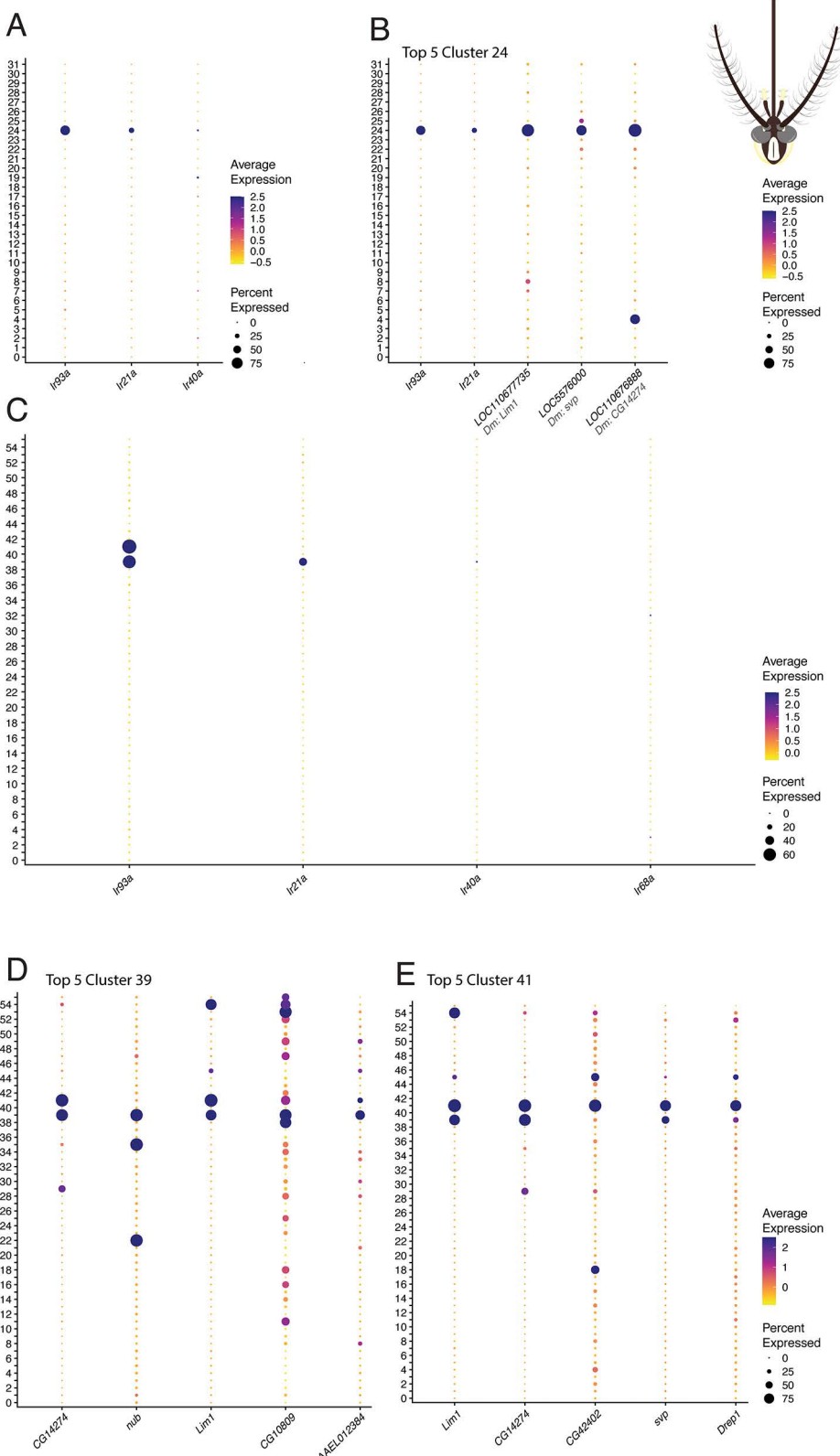

**Fig 2. HRN clusters identified in two independent *Ae. aegypti* antennal transcriptomes by co-expression of *Ir93a*, *Ir40a* and *Ir21a*.** Analysis of two independent datasets confirms reproducibility of HRN identification in *Ae. aegypti*. (A) Expression of *Ir93a*, *Ir40a* and *Ir21a* across antennal neuronal

clusters in the Herre et al.; *Ir68a* was not included as it was not significantly detected in this dataset. Cluster 24 shows specific co-expression of *Ir93a*, *Ir40a* and *Ir21a*, with *Ir40a* at notably lower levels than in *D. melanogaster*. (B) Top five marker genes for cluster 24; *D. melanogaster* orthologues are indicated in grey italics below each gene ID. (C) Expression of *Ir93a*, *Ir40a*, *Ir68a* and *Ir21a* across antennal neuronal clusters in the Adavi et al. dataset. Clusters 39 and 41 show co-expression of *Ir93a*, *Ir40a* and *Ir21a*; *Ir68a* expression is negligible. (D–E) Top five marker genes for clusters 39 and 41 respectively. Dot size represents the proportion of nuclei expressing the gene within that cluster (percent expressed); colour indicates average expression level, ranging from low (yellow) to high (purple).

arista cold, and hot cells, while noting its absence in dry cells. Within temperature-sensitive cells, including hygrocool cells in chambers I and II, as well as arista cold and hot cells, we found *nub* expression overlapping with *Ir93a*. Analysis of moist cells showed predominant expression of *salr* and *fred*. We detected the serotonin receptor *5-HT7* in dry and arista hot cells, while expression of *Kif19A* was specific to dry cells. The remaining genes displayed broader expression patterns across sacculus cells, suggesting they may function as more general markers for this neuronal population.

The identification of this set of conserved genes between *D. melanogaster* and *Ae. aegypti* sensory neurons provides insight into the core molecular components required for hygrosensation. The identified genes include diverse functional categories such as transcriptional regulators, signalling molecules, enzymes and structural proteins. This diversity suggests that humidity detection and response in dipterans rely on the coordinated activity of multiple conserved cellular pathways essential for hygrosensory function. While some of these genes have established roles in sensory systems, others represent novel candidates for future investigation. Their conservation across species, combined with their precise expression patterns in *D. melanogaster* sacculus neurons, indicates they likely play fundamental roles in how dipterans detect and process environmental humidity information.

**Loss of *5-HT7* and *Kif19A* impairs humidity-guided behaviour.** To investigate the functional significance of conserved genes in hygrosensation, we performed behavioural analyses in *D. melanogaster* mutants using a dynamic humidity arena. Desiccated and starved flies were exposed to step changes in relative humidity (10%→80%→10% RH), and locomotor speed modulation across these transitions was used as a readout of humidity-guided behaviour. Wild-type (*w¹¹¹⁸*) flies showed robust speed modulation, increasing locomotor activity under dry conditions (10% RH) and reducing activity under humid conditions (80% RH) (Fig 5A). From the conserved gene set, we tested genes where viable mutant lines were available, selecting *Ir21a*, *5-HT7* and *Kif19A* for behavioural analysis.

Despite its conserved expression in hygrocool neurons, *Ir21a* mutants exhibited normal humidity-driven locomotor responses indistinguishable from wild-type controls, with increased locomotor activity at low humidity and reduced activity at high humidity (Fig 5B). This result is consistent with previous studies and may reflect the restricted expression of *Ir21a* to hygrocool cells in chamber I only.

In contrast, loss of the serotonin receptor *5-HT7* resulted in a complete failure to modulate locomotor speed across humidity transitions, with mutant flies showing no significant difference in walking speed between dry and humid conditions (Fig 5C). Similarly, *Kif19A* mutants showed no significant speed modulation in response to humidity step changes (Fig 5D). Thus, both *5-HT7* and *Kif19A* are necessary for normal humidity-guided locomotor behaviour in *D. melanogaster*.

## Discussion

### Conserved genes in hygrosensation

Hygrosensation remains one of the few sensory modalities where the molecular transduction mechanism is not fully understood. Through our comparative analysis, we identified a conserved set of 21 genes associated with hygrosensory neuron function in *D. melanogaster* and *Ae. aegypti*. The conservation of these genes across dipteran species separated by over 150 million years of evolution suggests they represent fundamental components required for humidity sensing. These genes span multiple cellular functions, such as transcription factors, enzymes and membrane-proteins, suggesting humidity detection requires coordinated action across diverse molecular pathways.

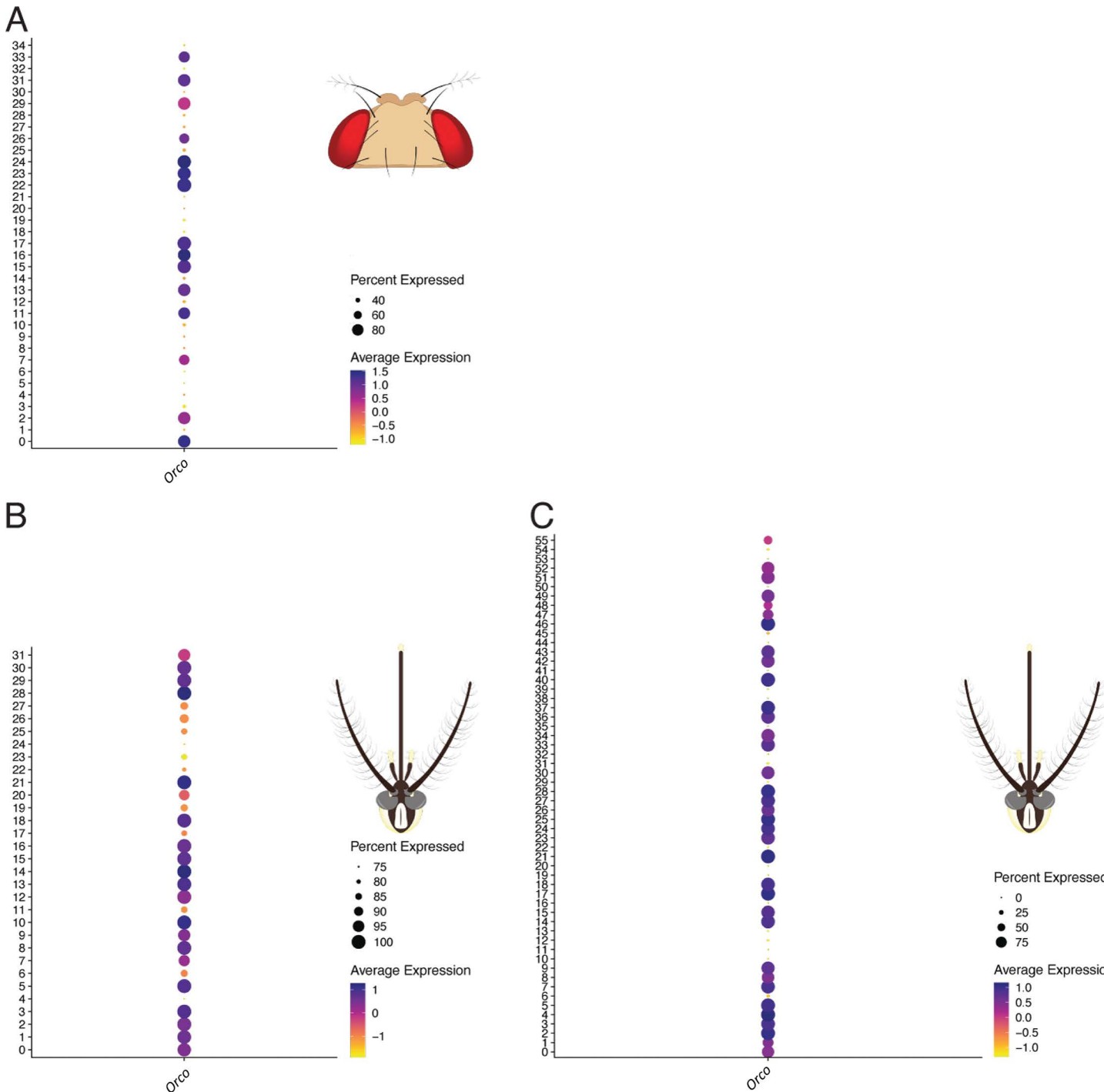

**Fig 3. Cross-species comparison reveals 21 shared marker genes in HRN clusters of *D. melanogaster* and *Ae. aegypti*.** (A) Expression of the 21 shared marker genes in HRN clusters of *D. melanogaster*. (B) Expression of shared marker genes in the *Ae. aegypti* HRN cluster Herre dataset; *D. melanogaster* orthologues indicated in grey italics. (C) Expression of shared marker genes in the two *Ae. aegypti* HRN clusters in the Adavi dataset. The size of each dot represents the proportion of nuclei expressing the gene within that cluster (percent expressed); colour indicates average expression level, ranging from low (yellow) to high (purple).

A

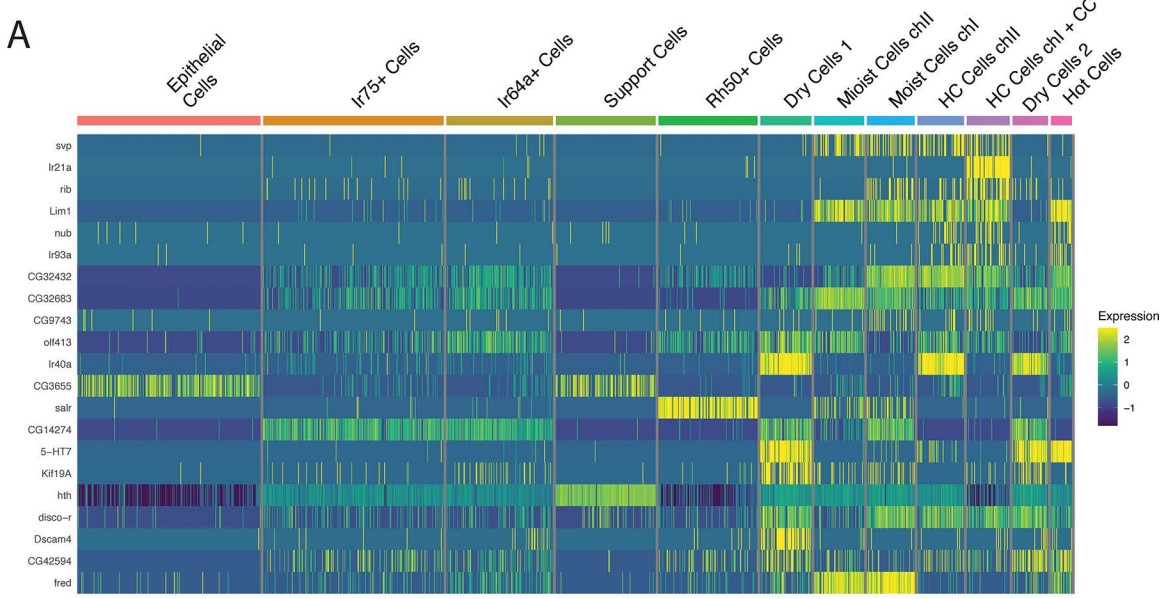

B

| D. melanogaster ENTREZ GENE | Ae. aegypti ENTREZ GENE | Ae. aegypti Vector Base |
|---|---|---|
| | | |
| svp | LOC5576000 | AAEL002765-svp |
| Ir21a | Ir21a | Ir21a |
| rib | LOC5568262 | AAEL006690-rib |
| Lim1 | LOC110677735 | AAEL019457-Lim1 |
| nub | LOC23687865 | AAEL017445-nub |
| Ir93a | Ir93a | Ir93a |
| CG32432 | LOC5574103 | AAEL025785-CG32432 |
| CG32683 | LOC5575719 | AAEL019587-CG32683 |
| CG9743 | LOC5578684 | AAEL003645-CG9743 |
| olf413 | LOC5578723 | AAEL026737-olf413 |
| Ir40a | Ir40a | Ir40a |
| CG3655 | LOC5565978 | AAEL027400-CG3655 |
| salr | LOC110677254 | AAEL021012-salr |
| CG14274 | LOC110676888 | AAEL025416-CG14274 |
| 5-HT7 | LOC5572158 | AAEL027242-5-HT7 |
| Kif19A | LOC5579633 | AAEL014134-Kif19A |
| hth | LOC5575090 | AAEL011643-hth |
| disco-r | LOC5577784 | AAEL013374-disco-r |
| Dscam4 | LOC5578262 | AAEL019773-Dscam4 |
| CG42594 | LOC5575382 | AAEL011786-CG42594 |
| fred | LOC5577413 | AAEL019702-ed |

**Fig 4. Subtype-specific expression of conserved hygrosensory genes across *D. melanogaster* antennal sacculus neuron subtypes and support cells.** (A) Heatmap showing scaled expression of genes conserved between *D. melanogaster* and *Ae. aegypti* HRN clusters, visualized across *D. melanogaster* antennal sacculus subtypes and support cells as previously described [25]. Columns correspond to: Epithelial cells, IR75a+ cell (olfactory neurons not in sacculus), Ir64a+ cells (acid-sensing, chamber III), Support Cells (glia), Rh50+ cells (ammonium-sensing, chamber III), Dry Cells 1 and 2 (see details in [25]), Moist Cells (chamber I and chamber II), Hygrocool Cells (HC, chamber II and chamber I; the latter also including arista Cold Cells, CC) and Hot Cells. Rows indicate conserved genes; colour indicates scaled expression level, ranging from low (blue) through mid-range (green) to high (yellow). (B) Table of *D. melanogaster* Entrez Gene identifiers and their corresponding *Ae. aegypti* Entrez Gene and VectorBase identifiers for the 21 conserved HRN marker genes.

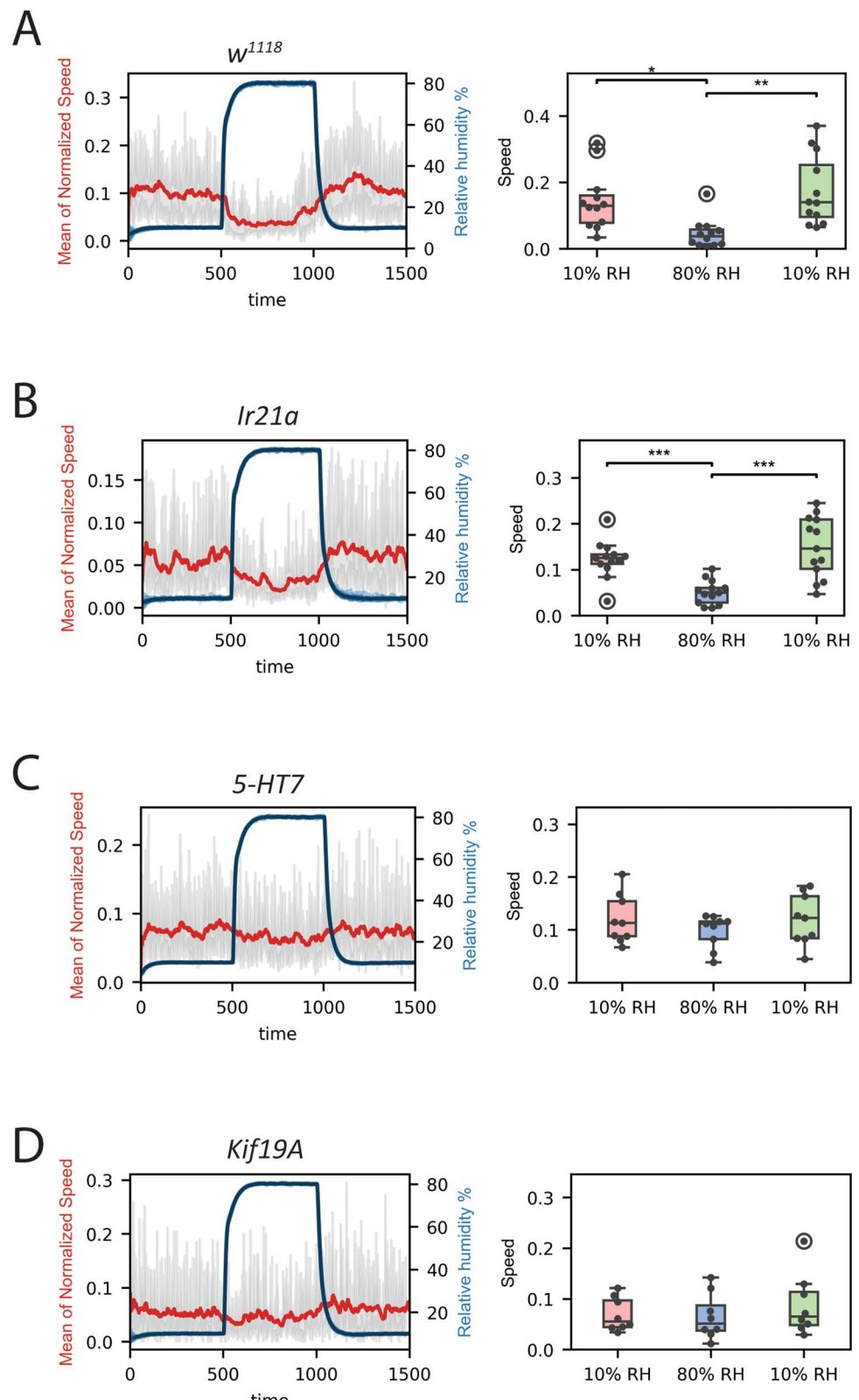

**Fig 5. Loss of *5-HT7* or *Kif19A*, but not *Ir21a*, impairs humidity-guided behaviour in *D. melanogaster*.** Behavioural responses of control and mutant flies to step changes in ambient relative humidity (RH) from 10% to 80% and back to 10%. (A) *w¹¹¹⁸* control strain (n = 13), showing increased

locomotor activity at low RH and decreased activity at high RH. (B) *Ir21a* mutant (n = 8), showing intact humidity-sensing behaviour comparable to controls. (C) *5-HT7* mutant (n = 10), showing significantly impaired humidity-sensing behaviour. (D) *Kif19A* mutant (n = 11), showing significantly impaired humidity-sensing behaviour. Boxplots summarize the distribution of fly-level responses, showing median (central line), interquartile range (box), upper and lower quartiles (whiskers), data from individual flies (dots) and statistical outliers (circled dots). Red lines show the mean normalized walking speed for each genotype over time, with grey shading representing the 95% confidence interval. Blue lines indicate real-time RH transitions during the assay. Asterisks denote levels of significance (*p < 0.05, **p < 0.01, ***p < 0.001).

Seven conserved transcription factors (*svp*, *rib*, *Lim1*, *nub*, *salr*, *hth* and *disco-r*) were identified and may form a regulatory framework that establishes and maintains hygrosensory neuron identity. Many of these factors have developmental roles. For example, *hth* is essential for proper antenna formation and segmental identity, while *Lim1* plays a critical role in specifying appendage development [37,38]. However, their continued expression in adult neurons suggests additional roles in maintaining sensory cell identity and function. The transcription factors show distinct expression patterns that align with specific sensory neuron subtypes: *salr* marks moist cells, while *nub* is expressed in temperature-sensitive neurons including hygrocool cells. The combinatorial organization of these transcription factors likely creates a regulatory code that defines the molecular and functional properties of each hygrosensory neuron type. The conservation of this transcriptional regulatory network between flies and mosquitoes suggests it has a key role in hygrosensory neuron differentiation and specification.

The cell adhesion molecules *fred* and *Dscam4* show distinct expression patterns in hygrosensory neurons, suggesting specialized roles in the structural organization among these cell types. *fred* coordinates cellular adhesion and signalling during development [39,40]. It is specifically expressed in moist cells where it may function to maintain their specialized cellular architecture within the sacculus. *Dscam4* is expressed in one subtype of dry cells, likely located to one chamber of the sacculus. Interestingly, previous work has shown that a different family member, *Dscam2*, marks dry cells in the other chamber, suggesting that different Dscam proteins help establish the distinct organization of dry cell subtypes [25]. Given that Dscam proteins mediate precise neuronal targeting through homophilic adhesion [41], this complementary expression pattern could be important for maintaining proper circuit architecture in the adult sacculus.

Two conserved GPI-anchored proteins, *CG14274/witty* and *CG32432*, potentially play important roles in organizing and regulating humidity receptor complexes. These proteins have functions in modulating ionotropic glutamate receptors (iGluRs) in other contexts: *CG14274* mediates receptor aggregation at synapses to ensure proper synaptic transmission [42], while the *C. elegans* ortholog of *CG32432*, called *sol-1*, acts as a transmembrane AMPAR regulatory protein (TARP) that stabilizes receptor conformations and influences gating properties [43,44]. Given the structural homology between IRs and iGluRs, we propose that CG14274/witty and CG32432 serve analogous roles in hygrosensory dendrites — organizing IR complexes and modulating their gating properties, respectively. Their conservation and specific expression in HRNs across both species make them strong candidates for further investigation.

The identification of these novel conserved genes between *D. melanogaster* and *Ae. aegypti* sensory neurons provides insight into the core molecular components required for hygrosensation. While some of these genes were previously known to function in sensory systems, others represent novel candidates for future investigation. Their conservation across species, combined with their precise expression patterns in *D. melanogaster* sacculus neurons, indicates they likely play fundamental roles in how dipterans detect and process environmental humidity information.

## Novel genes in hygrosensation

Our behavioural analysis of constitutive mutants suggests that *5-HT7* and *Kif19A* contribute to humidity-guided behaviour, potentially through their roles in HRNs where both genes show specific expression. In contrast, loss of *Ir21a* does not impair humidity-guided behaviour, consistent with previous studies [15,17,21]. As *Ir21a* is exclusively expressed in hygrocool cells of chamber I, these results reflect either a redundancy of *Ir21a* in these cells or a redundancy of these cells in the humidity-guided behaviours tested in these assays.

5-HT7 has previously been shown to coordinate sensory input with physiological responses in *D. melanogaster*, specifically in translating gustatory detection into changes in digestive function by activating enteric neurons to regulate crop contractions [45]. This fits with broader evidence that serotonergic systems can transform acute sensory detection into longer-term physiological changes, allowing animals to make anticipatory responses to environmental conditions [46]. We propose that serotonin, acting through 5-HT7, links humidity sensing to downstream physiological responses — a hypothesis consistent with serotonin#39;s established role as a diuretic hormone in several insect species [47]. Though whether humidity or desiccation state modulates serotonin levels in this context remains to be tested.

As the highest-affinity serotonin receptor in *D. melanogaster*, 5-HT7 is particularly well-suited to detect subtle changes in circulating serotonin levels, suggesting it could integrate both local and systemic serotonergic signals to modulate hygrosensory responses [48]. The dual role of 5-HT7 in both regulating feeding and humidity tracking also raises the possibility that these sensory modalities are coordinated to help flies maintain water balance, particularly important given that feeding state and humidity levels both impact desiccation risk. Further experiments with cell-specific manipulation will be needed to determine how 5-HT7 acts directly in the dry cells and how it modulates their activity.

The behavioural deficits observed in *Kif19A* mutants indicate that this kinesin may contribute to hygrosensory function, possibly through roles in intracellular transport or cilia maintenance, underscoring a potential link between cellular architecture and sensory neuron activity. Kif19A, a plus-end directed kinesin-8 family motor protein, exhibits microtubule-depolymerizing activity and localizes specifically to ciliary tips [49]. Studies in mice have demonstrated that loss of *Kif19A* leads to abnormally elongated cilia that cannot function properly [49]. Insect sensory cilia share the same fundamental primary cilium architecture (non-motile, 9+0) and conserved intraflagellar transport machinery as vertebrate sensory cilia [50]. As ciliary length and architecture is a critical factor of sensory function, Kif19A could have a role in controlling ciliary length in dry cells necessary for proper detection of humidity changes [51,52]. Furthermore, ciliary membrane composition and receptor localization are tightly regulated processes that are essential for sensory function [53]. As a plus-end directed motor, Kif19A may also play a role in trafficking sensory components, like IRs, to ciliary tips. However, the exact relationship between microtubule structure and hygrosensory function requires further investigation to fully understand the role of Kif19A in this sensory system.

These expanded roles for 5-HT7 and Kif19A highlight how receptor-mediated neuromodulation and specialized cellular transport may function to establish and maintain humidity sensing. The conservation of these components across species suggests they represent fundamental requirements for hygrosensory neuron function, providing new insights into the cellular machinery underlying environmental sensing in dipterans.

## Concluding remarks

Our integrative analysis of hygrosensory neurons across two phylogenetically distant dipteran species has revealed a conserved molecular toolkit essential for hygrosensation. By combining comparative transcriptomics with targeted behavioural analysis, we have uncovered multiple layers of cellular regulation in these specialized neurons. Beyond the previously characterized ionotropic receptors, we identified novel components spanning diverse cellular functions: the serotonin receptor *5-HT7*, suggesting potential neuromodulatory mechanisms, and the motor protein *Kif19A*, pointing to a role for specialized cellular transport. The conservation of these components between species separated by over 150 million years of evolution, together with their specific expression patterns in different sensory neuron subtypes, suggests fundamental organizational principles in HRNs. This approach, combining systematic genomic comparison with behavioural analysis, provides a framework for investigating sensory system evolution across species.

## Supporting information

**S1 Fig. Neuronal and glial cluster identification in the D. melanogaster antenna.** (A) Dot plot showing expression of the neuronal markers Syt1, elav, CadN and brp, and the glial marker repo, across all antennal clusters. (B) UMAP projection of cluster identities assigned as neuronal, non-neuronal or glia based on marker expression. Dot size represents the

proportion of nuclei expressing the gene (percent expressed); colour indicates average expression level, ranging from low (yellow) to high (purple).
(TIF)

**S2 Fig. Neuronal and glial cluster identification in the Ae. aegypti antenna (Herre et al., 2022).** (A) Dot plot showing expression of the neuronal markers LOC5565901 (orthologue to Syt1), LOC5570204 (orthologue to elav), LOC5564848 (orthologue to CadN) and LOC5570381 (orthologue to brp), and the glial marker LOC110678282 (orthologue to repo), across all antennal clusters. (B) UMAP projection of assigned cluster identities. Clusters 3 and 45 displayed ambiguous neuronal marker expression and were classified as ambiguous; all other clusters were assigned as neuronal, non-neuronal or glia. Dot size represents the proportion of nuclei expressing the gene (percent expressed); colour indicates average expression level, ranging from low (yellow) to high (purple).
(TIF)

**S3 Fig. Hygro- and thermoreceptor expression in the Ae. aegypti antennal dataset (Herre et al., 2022) confirms exclusion of clusters 3 and 45.** Dot plot showing expression of Ir25a, Ir93a, Ir40a and Ir21a across all neuronal clusters. Clusters 3 and 45 show negligible expression of Ir93a, Ir40a and Ir21a, supporting their exclusion from downstream hygrosensory analyses. Dot size represents the proportion of nuclei expressing the gene (percent expressed); colour indicates average expression level, ranging from low (yellow) to high (purple).
(TIF)

**S4 Fig. Absence of Orco expression in candidate HRN clusters confirms their non-olfactory identity.** Dot plots showing expression of Orco across all neuronal clusters in (A) D. melanogaster ((B) Ae. aegypti Herre et al. dataset and (C) Ae. aegypti Adavi et al. dataset. In all three datasets, the candidate HRN clusters (clusters 14 and 27 in D. melanogaster; cluster 24 in the Herre et al. dataset; clusters 39 and 41 in the Adavi et al. dataset) show negligible Orco expression, consistent with a hygrosensory rather than olfactory identity. Dot size represents the proportion of nuclei expressing the gene (percent expressed); colour indicates average expression level.
(TIF)

**S5 Fig. Top 50 marker genes for candidate HRN clusters in D. melanogaster.** Dot plots showing the top 50 marker genes identified by Seurat FindMarkers for (A) cluster 14 and (B) cluster 27 across all D. melanogaster antennal neuronal clusters. Each panel is displayed across two rows for readability. These marker gene lists were used as input for the cross-species conserved gene analysis. Dot size represents the proportion of nuclei expressing the gene within that cluster (percent expressed); colour indicates average expression level, ranging from low (yellow) to high (purple).
(TIF)

**S6 Fig. Top 50 marker genes for candidate HRN clusters in *Ae. aegypti*.** Dot plots showing the top 50 marker genes identified by Seurat FindMarkers for (A) cluster 24 (Herre et al.), (B) cluster 39 (Adavi et al.) and (C) cluster 41 (Adavi et al.) across all Ae. aegypti antennal neuronal clusters. Each panel is displayed across two rows for readability. D. melanogaster orthologues are indicated in grey italics. These marker gene lists were used as input for the cross-species conserved gene analysis. Dot size represents the proportion of nuclei expressing the gene within that cluster (percent expressed); colour indicates average expression level, ranging from low (yellow) to high (purple).
(TIF)

## Acknowledgments

We wish to thank the NBIS for their support on this project. The authors thank Marcus Stensmyr for comments on discussion of this project.

## Author contributions

**Conceptualization:** Kristina Corthals, Anders Enjin.

**Data curation:** Kristina Corthals, Ganesh Giri, Johan Reimegård, Allison Churcher.

**Formal analysis:** Kristina Corthals, Ganesh Giri.

**Funding acquisition:** Kristina Corthals.

**Investigation:** Kristina Corthals.

**Methodology:** Kristina Corthals.

**Resources:** Kristina Corthals.

**Software:** Kristina Corthals.

**Validation:** Kristina Corthals.

**Visualization:** Kristina Corthals.

**Writing – original draft:** Kristina Corthals, Anders Enjin.

**Writing – review & editing:** Ganesh Giri, Johan Reimegård, Allison Churcher.

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
