## [Decision Letter · Decision Letter 0]

13 Jan 2026

Dear Dr. Corthals,

plosone@plos.org. . . . A letter that responds to each point raised by the academic editor and reviewer(s). You should upload this letter as a separate file labeled 'Response to Reviewers'.A marked-up copy of your manuscript that highlights changes made to the original version. You should upload this as a separate file labeled 'Revised Manuscript with Track Changes'.An unmarked version of your revised paper without tracked changes. You should upload this as a separate file labeled 'Manuscript'.

We look forward to receiving your revised manuscript.

Kind regards,

Rachid Bouharroud

Academic Editor

PLOS One

Journal Requirements:

“Open access funding provided by Lund University. The funding was provided by Wenner-Gren Stiftelserna, Svenska Forskningsrådet Formas (2021-02008), Knut och Alice Wallenbergs Stiftelse, Vetenskapsrådet (202103772), Crafoordska Stiftelsen, Jeanssons Stiftelser. KC was financially supported by a Formas Mobility Grant (2021-02008).

AC and JR were financially supported by the SciLifeLab & Wallenberg Data Driven Life Science Program, Knut and Alice Wallenberg Foundation (KAW 2020.0239, KAW 2017.0003), and by the National Bioinformatics Infrastructure Sweden (NBIS) at SciLifeLab.”

Reviewers' comments:

Reviewer's Responses to Questions

**Comments to the Author**

1. Is the manuscript technically sound, and do the data support the conclusions?

Reviewer #1: Yes

Reviewer #2: Partly

2. Has the statistical analysis been performed appropriately and rigorously?

Reviewer #1: Yes

Reviewer #2: Yes

3. Have the authors made all data underlying the findings in their manuscript fully available?

Reviewer #1: Yes

Reviewer #2: Yes

4. Is the manuscript presented in an intelligible fashion and written in standard English?

Reviewer #1: Yes

Reviewer #2: Yes

Reviewer #1: The authors study uses single-nucleus transcriptomics and behavioral assays to identify conserved genes involved in humidity sensing in insects. By comparing Drosophila melanogaster and Ae. aegypti, the authors find a shared molecular signature in hygrosensory neurons across species separated by roughly 150 million years. Behavioral tests in flies show that three genes 5HT7, nubbin, and Kif19A are needed for normal humidity guided behavior. Overall, the work outlines a common molecular framework for how insects sense and respond to humidity.

Some of the comments and suggestions are as below

Are the behavioral defects specific to humidity sensing, or could they arise from broader sensory or locomotor issues?

Were proper controls included to rule out pleiotropic effects of genes like nubbin and 5 HT7?

It would help to briefly mention the type of humidity-guided behavior tested (e.g: preference, navigation, avoidance).

Is the “hygrocool neuron” widely accepted as part of the humidity-sensing pathway, or is it better viewed as a temperature related component within the hygrosensillum?

Consider acknowledging known nonIR pathways (GPCRs, mechanosensory channels, secnd messenger systems) to avoid an IRonly perspective.

The description of mutant phenotypes could be clearer.

If Ir21a mutants show normal humidity responses, what does this mean for redundancy or specialization among HRN subtypes?

Phrases like “define these neurons” could be softened to “contribute to” unless causality is demonstrated.

The sentence on lines 62–63 could be reworded for smoother flow.

Introduce abbreviations (e.g: HRNs) earlier and keep them consistent.

Gene name formatting should be uniform (e.g.: Ir40a, Ir68a).

Use the correct mosquito abbreviation: Ae. aegypti.

Materials and Methods

How many nuclei/cells remained after QC per dataset? A summary table with dataset size, depth, and QC filters (UMIs, mitochondrial cutoff) is needed.

With little or no Ir68a detected in either species, how was the moist-cell cluster confidently identified? Were other known markers used?

Clarify how HRN identity was verified when canonical markers were weak.

Bootstrap statistics are fine, but was multiple-testing correction applied?

Minor comments

Italicize gene names consistently.

Fix “Ae. aegpyti” to Ae. aegypti.

Break long paragraphs (89–100, 151–159) for clarity.

“Transcriptome analysis” to “Single-nucleus transcriptomic analysis.”

Results

Clarify whether cluster 14 was treated as a combined HRN group or whether subclustering was done.

Are there nonHRN cell types that express Ir40a in mosquitoes, or is expression HRN specific?

State the number of flies tested per genotype and the variability across individuals.

Re state whether behavioral deficits could reflect general locomotor or motivational issues.

Conclusions about “functional conservation” should be softened because functional tests were only done in Drosophila. Emphasize transcriptional conservation for mosquitoes.

minor commnets

Keep gene formatting consistent (Kif19A vs. Kif19a).

Add brief one-sentence summaries of figures.

Avoid strong mechanistic claims without direct evidence.

Fix spacing, capitalization, and typographical errors.

Discussion

Clarify that conservation across dipterans is shown at the transcriptional level; direct functional conservation is only tested in flies.

The “regulatory code” of seven transcription factors should be presented as a proposed model, not a confirmed network.

For fred and Dscam4, note that the suggested role is based on expression and known Dscam functions, not direct anatomical evidence.

The analogy between iGluR modulators and IR-associated GPI-anchored proteins should be stated as a hypothesis without current biochemical proof.

The serotonin link is interesting, frame it as a testable idea since changes in serotonin with humidity/desiccation have not been shown.

Briefly clarify how insect sensory cilia compare to vertebrate cilia, since the discussion relies on this parallel.

Minor comments

Separate Ir21a temperature vs. humidity roles more clearly.

Correct merged citations around lines 349–352.

Remove repeated points between lines 336–342 and earlier.

Keep gene name formatting consistent.

Figures

Fig 2: Use Ae. aegypti consistently.

Add a note that two independent datasets were analyzed for reproducibility.

Fig. 3: Reduce method details in the legend; move OrthoDB methods to the main Methods section. Add a simple concluding sentence about the conserved HRN signature.

Fig. 4: Fix: “Hygrocool Cells (HC, chamber I/I)” → “chamber I/II.”

Fix spacing in “Rh50+ (ammonium...)”.

Keep gene formatting consistent; define all abbreviations at first use.

Move extra method details out of legends where possible.

Reviewer #2: PONE-D-25-60937_reviewer_comments:

Overall comments:

All terrestrial animals, including insects, need to be able to sense humidity in their environment to coordinate internal homeostasis necessary for survival. Sensing humidity involves specialized humidity receptor neurons (HRNs), which have been studied elsewhere with many key players identified, but the full picture remains elusive. In this current study, the authors compared humidity sensing neurons in two dipteran insects, the disease vector mosquito Aedes aegypti, and the model organism, Drosophila melanogaster. The authors identified 21 novel genes that appear to be shared in HRNs within these two organisms and may facilitate hygroreceptor functioning. Of these 21 candidates, the authors pursued further behavioural studies on three of these genes, using available mutant lines in Drosophila. However, these were mutant lines with ubiquitous deficiency - mutants affect the entire fly, not only in the HRNs where the authors demonstrated these genes are enriched. While the overall manuscript is well organized and written, the authors could strengthen this study by using additional mutant lines for the same target genes and also carry out specific knockdown/knockout in HRNs to confirm that the changes in humidity-responsive behaviour are indeed a result of expression of these three candidate genes (specifically 5-HT7, nubbin and Kif19A) in the HRNs. Otherwise, the authors can clarify that the data suggests generally the involvement of these genes in humidity responsive behaviours and not with certainty their expression in the HRNs. Also, since the authors carried out a comparative study on two dipteran insects, any extrapolation to additional insects beyond dipterans is highly speculative and should be removed.

Specific and generally minor comments:

Abstract

L25-27: The broad claim here in the abstract that the current data suggests shared functional requirements for hygrosenation in insects is unjustified given the current study focuses on only two dipteran species, which are not representative of all insects. Infact, in all other sections of the manuscript, the authors are indeed more conservative with their conclusions indicating these genes likely play fundamental roles in hygrosensation in dipterans.

Introduction

L61: remove “and”

Methods

Are public data sets in flies based on female flies? Authors could clarify since they are comparing to female mosquito datasets.

What about in extractions collected for de novo analysis in the current study?

L104 and L108: no problem with how identity of neuronal and glial cells was assigned but authors should add appropriate references in support of these designations.

L110: typo

L124-135: These two small paragraphs are redundant and should be combined. Also, authors should provide references for IRs previously described in hygrosensory neurons.

L161-164: Authors should clarify the function/purpose of each of the mutant strains. Additionally, are conclusions drawn on single mutant lines? Were lines backcrossed with controls? If not, shouldn’t they be?

Results

In results section (and elsewhere), all genes/transcripts should be written following standardized nomenclature, including italics to differentiate from proteins (see HUGO guidelines: https://pmc.ncbi.nlm.nih.gov/articles/PMC7494048/

or FlyBase: https://wiki.flybase.org/wiki/FlyBase:Nomenclature).

L214: earlier in this section the authors indicated three clusters were identified expressing Ir93a, Ir40a and Ir21a that were also negative for olfactory neuron markers, but didn’t see any text describing details of the last one, cluster 32.

L219: typo, italics

L272-277: with each of the available (single) mutant lines used, how can the authors confirm that loss of each gene (lr21a, 5-HT7, and Kif19a; or hypomorpohic nub) specifically in hygrosensory structures explains the changes in humidity-driven behaviour? This conclusion would be strengthened by both independent mutants and cell specific knockdown/knockout in each of the different chamber cells. Otherwise, given the complexity in generating behaviour including input, integration and output, how can authors ascertain that their data shows a direct action involving only the hygrosensory neurons?

Discussion

L297 and onwards: see comments raised earlier regarding gene nomenclature.

L314: typo in citation.

L341-342: this sentence suffers from the same issue as in the abstract. Now the authors are extending their claims onto insects more broadly with no evidence beyond these two dipterans. The sentence should be revised given no evidence beyond these two species is provided in the current dataset, nor is there any reference to available evidence in the literature. The sentence is also missing a period.

L397: this final sentence should, again, be more specific given the focus on two dipteran insects and not insects in general.

L402-403: without the additional independent mutant lines and cell-specific knockdown/knockout, this statement is not specifically supported by the data since the mutants were global and not confined to the hygrosensory neurons. Thus, while expression of these candidates genes in hygrosensation were confirmed in the different neuronal clusters, their specific requirement in these neurons was not validated (which would support the statement provided by the authors in this section).

.

Reviewer #1: **Yes:** Vinaya ShettyVinaya ShettyVinaya ShettyVinaya Shetty

Reviewer #2: No

---

## [Author Response · Author response to Decision Letter 1]

27 Mar 2026

Response to Reviewers and Editor Comments

PLOS ONE Manuscript ID: PONE-D-25-60937

Dear Editor and Reviewers,

We thank you for your constructive feedback on our manuscript. We have carefully considered all comments and have made substantial revisions to address the concerns raised. Most notably, we have removed the nubbin behavioral data in response to concerns about the hypomorphic nature of the mutant allele and the limitations of constitutive mutants for establishing cell-autonomous functions. We have also systematically revised the manuscript to limit our conclusions to dipteran insects rather than making broader claims about insects in general, and have explicitly acknowledged the limitations of our approach where appropriate.

Below, we provide detailed point-by-point responses to each comment. Editor and reviewer comments are shown in red text, and our responses follow in regular text.

EDITOR REQUIREMENTS

Corrected. We have ensured the manuscript meets all PLOS ONE style requirements.

Corrected. We have removed the funding statement from the manuscript body. The funding information is provided only in the online submission form.

3. We note that the grant information you provided in the 'Funding Information' and 'Financial Disclosure' sections do not match. When you resubmit, please ensure that you provide the correct grant numbers for the awards you received for your study in the 'Funding Information' section.

Corrected.

5. When completing the data availability statement of the submission form, you indicated that you will make your data available on acceptance. We strongly recommend all authors decide on a data sharing plan before acceptance, as the process can be lengthy and hold up publication timelines.

Clarified. All data are publicly available. The D. melanogaster dataset is from Li et al. (2022) Fly Cell Atlas (Science 375:eabk2432). The Ae. aegypti datasets are from Herre et al. (2022, Cell 185:3104-3123.e28) and Adavi et al. (2024, bioRxiv: 2024.08.21.608847). Analysis code is available at https://github.com/hygrosensation/ComparativeStudy.

6. Please include captions for your Supporting Information files at the end of your manuscript, and update any in-text citations to match accordingly.

Captions for all Supporting Information figures (S1–S6 Fig) have been added at the end of the revised manuscript following PLOS ONE guidelines, with corresponding in-text citations updated throughout.

7. Please review your reference list to ensure that it is complete and correct. If you have cited papers that have been retracted, please include the rationale for doing so in the manuscript text, or remove these references and replace them with relevant current references.

Fixed.

REVIEWER #1

General Assessment: The authors study uses single-nucleus transcriptomics and behavioral assays to identify conserved genes involved in humidity sensing in insects. By comparing Drosophila melanogaster and Ae. aegypti, the authors find a shared molecular signature in hygrosensory neurons across species separated by roughly 150 million years. Behavioral tests in flies show that three genes 5HT7, nubbin, and Kif19A are needed for normal humidity guided behavior. Overall, the work outlines a common molecular framework for how insects sense and respond to humidity.

We thank the reviewer for this positive assessment of our work.

Are the behavioral defects specific to humidity sensing, or could they arise from broader sensory or locomotor issues? Were proper controls included to rule out pleiotropic effects of genes like nubbin and 5 HT7?

We acknowledge that without cell-type-specific manipulations, broader pleiotropic effects cannot be definitively excluded. However, several considerations argue against a general locomotor or sensory deficit as the explanation for the observed phenotypes.

The behavioral paradigm itself dissociates general locomotor capacity from humidity-specific speed modulation. As demonstrated in Giri et al. (2024), desiccated Ir93a mutants tested in the same assay show normal overall walking activity and total distance covered compared to desiccated controls yet completely fail to modulate walking speed in response to humidity transitions. This establishes that the assay is selectively sensitive to hygrosensory impairment rather than general locomotor or motivational deficits — a fly with non-specific motor problems would show reduced locomotion across all conditions rather than a selective failure to respond to humidity transitions. The phenotypes observed in 5-HT7 and Kif19A mutants are therefore most reasonably interpreted as reflecting impaired humidity sensing.

It would help to briefly mention the type of humidity-guided behavior tested (e.g: preference, navigation, avoidance).

We have expanded the results section discussing the behavior experiments to make it clearer for the reader.

Is the "hygrocool neuron" widely accepted as part of the humidity-sensing pathway, or is it better viewed as a temperature-related component within the hygrosensillum?

Hygrocool neurons are canonically defined as members of the hygrosensory triad and are present in hygrosensilla of all insects studied (Altner, H., and R. Loftus. Ann Rev Entom 30.1 (1985): 273-295). While they respond to temperature stimuli and not humidity, they remain integral to the hygrosensillum structure and share expression of core receptors (Ir93a, Ir25a) with humidity-sensing neurons. Their exact function in hygrosensation is unknown.

Consider acknowledging known nonIR pathways (GPCRs, mechanosensory channels, second messenger systems) to avoid an IRonly perspective.

We thank the reviewer for this suggestion. We have added a sentence to the Introduction acknowledging that additional molecular components beyond ionotropic receptors have been implicated in hygrosensation, including the TRP channels water witch and nanchung, and the odorant binding protein Obp59a. We note that while water witch and nanchung have been associated with humidity-related behaviours, they have not been shown to be expressed in HRNs, and their precise role in hygrosensation remains unclear and both our group and others have not seen a hygrosensory phenotype in mutants of these genes (Enjin et al. 2016, Knecht et al. 2016). Obp59a is expressed in support cells of hygrosensory sensilla rather than in HRNs themselves, highlighting that the molecular apparatus underlying humidity detection extends beyond sensory receptor proteins.

The description of mutant phenotypes could be clearer.

We have expanded the results section to make this clearer to the reader.

If Ir21a mutants show normal humidity responses, what does this mean for redundancy or specialization among HRN subtypes?

We have added a brief clarification to the Discussion addressing this point. Ir21a is exclusively expressed in hygrocool cells of chamber I, with moist cells and dry cells of both chambers and hygrocool cells of chamber II are all Ir21a-negative. The normal humidity-guided behaviour of Ir21a mutants therefore either reflects functional redundancy within hygrocool chamber I cells specifically or indicates that these cells are not strictly required for the humidity step-change responses measured in our assay. The new discussion now reads: “In contrast, loss of Ir21a does not impair humidity-guided behaviour, consistent with previous studies [15,17,21]. As Ir21a is exclusively expressed in hygrocool cells of chamber I, these results reflect either a redundancy of Ir21a in these cells or a redundancy of these cells in the humidity-guided behaviors tested in these assays.”

Phrases like "define these neurons" could be softened to "contribute to" unless causality is demonstrated.

Corrected. We have revised language throughout to be more cautious.

The sentence on lines 62--63 could be reworded for smoother flow.

Sentence changed to “These ionotropic receptor-mediated mechanisms are conserved in mosquitoes”.

Introduce abbreviations (e.g: HRNs) earlier and keep them consistent.

Corrected. We have introduced HRN (humidity receptor neurons) at its first use in the Abstract and maintain consistent usage throughout the manuscript.

Gene name formatting should be uniform (e.g.: Ir40a, Ir68a).

Corrected. We have systematically checked and corrected gene name formatting throughout the manuscript to ensure consistency (italicized gene names, non-italicized protein names where appropriate).

Use the correct mosquito abbreviation: Ae. aegypti.

Corrected. We have corrected all instances of "Ae. aegpyti" to "Ae. aegypti".

How many nuclei/cells remained after QC per dataset? A summary table with dataset size, depth, and QC filters (UMIs, mitochondrial cutoff) is needed.

The mosquito antennal snRNA-seq datasets were obtained from Herre et al. (2022) and Adavi et al. (2024), where comprehensive QC metrics (dataset size, depth, UMI counts, mitochondrial cutoffs) are fully reported in the original publications. Similarly, QC details for the Drosophila dataset were reported in our previous work (Corthals et al., 2023). In response to your request, we have now added a summary in the Methods section (lines 129ff) specifying the number of cells retained from each dataset after our neuron-specific filtering, ensuring clarity regarding the cell numbers used in our hygrosensory analyses.

With little or no Ir68a detected in either species, how was the moist-cell cluster confidently identified? Were other known markers used?

Moist cells were identified based on expression of previously characterized markers from our earlier transcriptomic study (Corthals et al., 2023), including Ac13E, Nlg3, and fred, rather than relying solely on Ir68a. The low detection of Ir68a in single-nucleus RNA-seq is consistent with previous studies (Corthals et al., 2023, McLaughlin et al., eLife 10:e63856) and may reflect technical limitations of the method for detecting lowly expressed genes or genuinely low expression levels in the adult antenna.

Clarify how HRN identity was verified when canonical markers were weak.

We identified HRN clusters based on the combination of: (1) expression of Ir93a, Ir40a, and Ir21a, which are established hygro- and thermoreceptors; (2) absence of olfactory neuron markers (Orco, Supplementary Figure 4); and (3) cross-referencing with our previous detailed transcriptomic analysis of the sacculus (Corthals et al., 2023). In Ae. aegypti, the presence of Ir40a, which is exclusively expressed in HRNs based on functional studies (Tang et al., 2024), combined with consistent expression of Ir93a and Ir21a across both independent datasets (Herre et al., 2022; Adavi et al., 2024), established cluster identity with high confidence.

Bootstrap statistics are fine, but was multiple-testing correction applied?

We thank the reviewer for this important point. In the original analysis, explicit multiple-testing correction on the bootstrap-derived p values was indeed missing. We have now re-analysed the data and updated the corresponding figure accordingly. The overall conclusions of the behavioural analysis remain unchanged.

"Transcriptome analysis" to "Single-nucleus transcriptomic analysis."

Corrected.

Clarify whether cluster 14 was treated as a combined HRN group or whether subclustering was done.

Cluster 14 was analyzed as a combined HRN group without subclustering. Given the limited number of hygrosensory receptor neurons (HRNs) typically present in a single Drosophila melanogaster antenna, we chose not to pursue subclustering to avoid risking the loss of HRN neurons. This has been clarified in the results section of the manuscript.

Are there nonHRN cell types that express Ir40a in mosquitoes, or is expression HRN specific?

As described by Tang et al (2024), Ir40a expression in Ae. aegypti labels dry cells in sensilla ampullacea and coeloconic sensilla (flagellomeres 10/12), alongside thermosensitive neurons in tip coeloconic sensilla (flagellomere 13). This distribution parallels D. melanogaster, where Ir40a marks dry cells in both chambers and hygrocool cells in chamber I, the latter have been shown to integrate temperature/humidity signals. As the used single-nucleus datasets are restricted to antennal neurons, it excludes the minor population of internal Ir40a+ neurons near the cibarium. This indicates that Ir40a expression appropriately identifies HRNs in the Ae. aegypti antenna.

State the number of flies tested per genotype and the variability across individuals.

Sample sizes are now provided in the Figure 5 legend: w1118 control (n=13), Ir21a (n=8), 5-HT7 (n=10 ), and Kif19A (n=11) individual flies. Individual fly responses are shown as dots on the box plots, allowing assessment of between-individual variability.

Restate whether behavioral deficits could reflect general locomotor or motivational issues.

In the behavioral assay used, flies typically maintain a consistent walking speed under low humidity conditions (10% RH) and slow down or stop upon encountering high humidity (80% RH) (see details in Giri et al., 2024). In our experiments, all tested fly strains, except nub, reached comparable walking speeds at 10% RH. All data discussing nub has therefore been removed. The consistency of the other strains indicates that baseline locomotor activity and motivation were unaffected in this setup. We therefore conclude that the observed behavioral phenotypes are specific to altered hygrosensory processing rather than reflecting general locomotor or motivational impairments.

Conclusions about "functional conservation" should be softened because functional tests were only done in Drosophila. Emphasize transcriptional conservation for mosquitoes.

Corrected.

Keep gene formatting consistent (Kif19A vs. Kif19a).

Corrected. We have standardized Kif19A throughout the manuscript, following the standard nomenclature.

Add brief one-sentence summaries of figures.

Corrected. We have reviewed all figure legends and ensured each begins with a clear summary sentence describing the main finding of the figure .

Avoid strong mechanistic claims without direct evidence.

Corrected. We have softened language throughout, particularly in the Discussion, to frame mechanistic proposals as hypotheses rather than established mechanisms.

Fix spacing, capitalization, and typographical errors.

Corrected. We have carefully proofread the manuscript and corrected typographical errors.

Clarify that conservation across dipterans is shown at the transcriptional level; direct functional conservation is only tested in flies.

Corrected. We have clarified this throughout the Discussion. The conservation analysis is explicitly described as transcriptomic/molecular, while behavioral validation is limited to D. melanogaster. We consistently use cautious language about potential roles in mosquitoes, stating that functional studies in Ae. aegypti would be needed to confirm conserved roles.

The "regulatory code" of seven transcription factors should be presented as a proposed model, not a confirmed network.

Corrected. We have revised to “Seven conserved transcription factors (svp, rib, Lim1, nub, salr, hth and disco-r) were identified and may form a regulatory framework that establishes and maintains hygrosensory neuron identity”

For fred and Dscam4, note that the suggested role is based on expression and known Dscam functions, not direct anatomical evidence.

Corrected.

The analogy between iGluR modulators and IR-associated GPI-anchored proteins should be stated

---

## [Editor Report · Decision Letter 1]

9 Apr 2026

Conserved molecular signatures of hygrosensory neurons in two dipteran species

PONE-D-25-60937R1

Dear Dr. Corthals,

We’re pleased to inform you that your manuscript has been judged scientifically suitable for publication and will be formally accepted for publication once it meets all outstanding technical requirements.

Kind regards,

Rachid Bouharroud

Academic Editor

PLOS One
---

## [Editor Report · Acceptance letter]

PONE-D-25-60937R1

PLOS One

Dear Dr. Corthals,

I'm pleased to inform you that your manuscript has been deemed suitable for publication in PLOS One. Congratulations! Your manuscript is now being handed over to our production team.

Kind regards,

on behalf of

Dr. Rachid Bouharroud

Academic Editor

PLOS One